# Selection of reference genes for quantitative analysis of microRNA expression in three different types of cancer

Yuliya A. Veryaskina[1,2]*, Sergei E. Titov[2,3], Mikhail K. Ivanov[3], Pavel S. Ruzankin[4,5], Anton S. Tarasenko[4,5], Sergei P. Shevchenko[6], Igor B. Kovynev[7], Evgenij V. Stupak[8], Tatiana I. Pospelova[7], Igor F. Zhimulev[2]

1 Laboratory of Gene Engineering, Institute of Cytology and Genetics, Siberian Branch of Russian Academy of Sciences, Novosibirsk, Russia, 2 Department of the Structure and Function of Chromosomes, Laboratory of Molecular Genetics Institute of Molecular and Cellular Biology, Siberian Branch of Russian Academy of Sciences, Novosibirsk, Russia, 3 AO Vector-Best, Novosibirsk, Russia, 4 Sobolev Institute of Mathematics, Siberian Branch of Russian Academy of Sciences, Novosibirsk, Russia, 5 Department of Mathematics and Mechanics, Novosibirsk State University, Novosibirsk, Russia, 6 Novosibirsk Municipal Clinical Hospital #1, Novosibirsk, Russia, 7 Department of Therapy, Hematology and Transfusiology, Novosibirsk State Medical University, Novosibirsk, Russia, 8 Department of Neurosurgery, Ya.L. Tsivyan Novosibirsk Research Institute of Traumatology and Orthopedics, Novosibirsk, Russia

* microrna@inbox.ru

**Data Availability Statement:** All relevant data are within the paper and its Supporting Information files.

## Abstract

MicroRNAs (miRNAs) are promising biomarkers in cancer research. Quantitative PCR (qPCR), also known as real-time PCR, is the most frequently used technique for measuring miRNA expression levels. The use of this technique, however, requires that expression data be normalized against reference genes. The problem is that a universal internal control for quantitative analysis of miRNA expression by qPCR has yet to be known. The aim of this work was to find the miRNAs with stable expression in the thyroid gland, brain and bone marrow according to NanoString nCounter miRNA quantification data. As a results, the most stably expressed miRNAs were as follows: miR-361-3p, -151a-3p and -29b-3p in the thyroid gland; miR-15a-5p, -194-5p and -532-5p in the brain; miR-140-5p, -148b-3p and -362-5p in bone marrow; and miR-423-5p, -28-5p and -532-5p, no matter what tissue type. These miRNAs represent promising reference genes for miRNA quantification by qPCR.

## Introduction

The discovery of the small non-coding RNA lin-4 in Caenorhabditis elegans in 1993 established a base for a new line of research in molecular biology [1]. 2,654 human miRNA genes are currently deposited in the miRBase database [2]. However, some of them are false-positive entries [3]. This led to the need of alternative databases that could give a true view of all miRNAs in existence. One of such databases is MirGeneDB 2.0: in 2020, it contained 556 validated human miRNAs [4]. MicroRNAs play an important regulatory role in many organisms in many biological processes. There is little doubt that aberrant miRNA expression may entail disease initiation and progression [5].

**Funding:** The work of S. Titov was financially supported by the Russian Science Foundation (project No. 20-14-00074). The work of Y. Veryaskina was supported by the Russian Foundation for Basic Research (project No. 19-34-60024). The work of P. Ruzankin and A. Tarasenko was supported by the Mathematical Center in Akademgorodok under agreement No. 075-15-2019-1675 with the Ministry of Science and Higher Education of the Russian Federation. The funders had no role in study design, data collection and analysis, decision to publish, or preparation of the manuscript.

**Competing interests:** The authors have declared that no competing interests exist.

To date, a large number of works showing how miRNAs are distributed in healthy and diseased human tissues have been published [6, 7]. These studies normally seek to identify human disease-specific miRNAs that have promise for personalized medicine, hence the need to have accurate miRNA profiling data on the study samples. The most commonly used approach to addressing this problem is through analysis of miRNA expression levels by qPCR [8]. This technique is highly sensitive, reproducible and relatively cheap. The main problem with qPCR-based protocols is one of the choice of an optimum reference gene. Normalization compensates for variations in the amounts of miRNA following RNA extraction, reverse transcription or that are due to variations in amplification efficiency. A reference gene should have stable expression in all samples, be expressed together with its target in the study cells and have a similar stability with target miRNAs, no matter what sample storage conditions. However, we have yet to know a universal reference gene meeting all these criteria [9]. It has been argued that normalization is the most accurate, if the reference genes used are in the same class of RNA as the study targets [10]. Nevertheless, miRNA expression studies most commonly use non-coding RNAs (U6, U48, U44 etc.). However, non-coding RNAs are not in the same class of RNA as miRNAs and, therefore, the properties of the former are not the same as those of the latter, which may results in different efficiencies of extraction, reverse transcription and PCR amplification. Various authors have demonstrated that, along with miRNAs, small nuclear RNAs are highly variable in expression, too [11–13]. Multiple studies have demonstrated that the best choice of reference gene in the analysis of the relative miRNA expression level is either a miRNA or the geometric mean of several stably expressed miRNAs. Additionally, it is recommended that a reference gene be searched for in each experimental system [9]. The most popular algorithms used for expression stability assessment are geNorm, NormFinder and Best Keeper [14–16].

Although qPCR is the most popular technique for measuring miRNA expression levels, it is not the only one: other molecular genetic methods that can assess the absolute number of miRNA molecules from one to hundreds at once are in use, too, for example, digital PCR, NanoString nCounter and sequencing [17–19]. One of the most popular multiplexed systems used for measuring gene expression is NanoString nCounter. This technique relies on digital barcoding, requires neither mRNA-to-cDNA conversion by reverse transcription nor cDNA amplification by PCR [20]. One experimental session included analysis of about 800 miRNAs in 12 study samples, showing that some miRNAs have variable expression, while the others, stable. Apparently, the miRNA identified by this technique as being stably expressed may further be used as reference genes for qPCR.

The aim of this work was to find the miRNAs that could be used as potential reference genes from among those miRNAs that are stably represented in the thyroid gland, brain and bone marrow, according to nCounter miRNA Assay data.

## Materials and methods

### Clinical samples

**1. Thyroid tissue.** All biological material was obtained in compliance with the legislation of the Russian Federation, and written informed consent was provided by all the patients, all the data were depersonalized. This study was approved by the ethics committee of the Institute of Molecular Biology and Biophysics of the Siberian Branch of the Russian Academy of Medical Sciences. We used samples of thyroid tumor tissue surgically removed from 32 patients and representing different neoplasm histotypes and normal tissue: normal tissue, 16; hyperplastic nodule, 5; papillary thyroid carcinoma, 2; follicular variant of papillary thyroid carcinoma, 2; follicular thyroid adenoma, 6; and follicular thyroid carcinoma, 1. Sample collection and

histology analysis were controlled by a qualified oncologist (Novosibirsk Municipal Clinical Hospital #1, Oncology Department VI). Detailed clinical information of the patients is given in S1 Table.

**2. Brain tissue.** Surgery material was two 1-mm$^3$ bioptic samples from each patient's brain: tumoral tissue and non-tumoral tissue taken farther than 2 cm away from the tumor margin, in a functionally insignificant location. Tumor samples were examined by histology and assigned to groups according to the 2007 WHO grading system of gliomas: Grade I (n = 1), Grade II (n = 2), Grade II (n = 2) and Grade IV (n = 1). Detailed clinical information of the patients is given in S2 Table.

**3. Bone marrow tissue.** We used 12 cytological specimens obtained by bone marrow aspiration in the Municipal Hematological Center of the Ministry of Health of the Novosibirsk Region. The study groups were patients with myelodysplastic syndrome (MDS) (n = 3), non-Hodgkin's lymphoma without myelodysplasia ((NHL(−MD)) (n = 3) and NHL with myelodysplasia ((NHL(+MD)) (n = 3); the control group consisted of patients with non-cancerous blood diseases (NCBD) (n = 3). Work with healthy bone marrow donors for allogeneic transplantation is beyond the competence of our clinic. Taken together, we decided to go with a control group composed of people who had no hematologic cancer, but had indications for bone marrow examination to exclude one. They were people with secondary anemic and cytopenic conditions, in whom leukemias were not confirmed by myelography. Detailed clinical information of the patients is given in S3 Table.

## Isolation of total RNA

Total RNA was extracted using the RNeasy Mini kit (QIAGEN, Valencia, CA, USA) according to the manufacture's recommendations.

## NanoString nCounter miRNA Expression Assay for miRNA profiling

The expression of 800 miRNAs in tumoral and normal thyroid and brain tissues was evaluated using the nCounter Human v2 miRNA Expression Assay Kit (NanoString Technologies, Seattle, WA, USA); and that in bone-marrow tissue, using the nCounter Human v3 miRNA Expression Assay Kit (NanoString Technologies, Inc., Seattle, WA, USA). These procedures were carried out in accordance with the manufacturer's protocol. For the NanoString assay, 100 ng of total RNA was isolated from bone-marrow aspiration material. The RNA concentration was measured by a NanoDrop 2000 spectrophotometer (Thermo Fisher Scientific, Inc., Waltham, MA, USA). According to NanoString recommendations, the 260/280 ratio should be not less than 1.9 and the 260/230 ratio, not less than 1.8, and so were our figures. The data were analyzed using the nSolver v4 package (NanoString Technologies, Inc., Seattle, WA, USA). The nCounter assay for each sample consisted of six positive controls, eight negative controls and five control mRNAs (ACTB, B2M, GAPDH, RPL19, RPLP0). Probes that recognize synthetic mRNA targets were included in the CodeSet at specified concentrations.

## Statistical analysis

The following groups of samples were considered for analysis of miRNA expression: 'thyroid gland', 'gliomas', 'bone marrow', and 'pooled'. In each group, the miRNAs that revealed less than 50 molecules in more than 50% of the samples were excluded from analysis. After normalization, the within-group variation of the binary logarithm of the normalized miRNA concentration was analyzed.

For each group, the following normalization strategies were considered: normalization using housekeeping genes; normalization using all miRNAs found; normalization using 75

most represented miRNAs; and normalization using the positive control. Normalized data were calculated as the difference between the binary logarithm of the registered number of miRNA molecules and the binary logarithm of the total number of normalizing molecules.

We used the following measures of variability for the normalized observation data: standard deviation, range (the difference between the maximum and the minimum), interquartile range (the difference between the third quartile and the first quartile) and the mean absolute deviation from the median. For each normalization strategy and each measure of variability, 10 least variable miRNAs are presented in S4–S7 Tables.

Additionally, we determined the most stable miRNAs according to all these measures of variability and two normalization strategies: one to housekeeping genes and one to the total miRNA content. Because we were not aware whether some of the measures of variability or some of the normalization strategies might be more important than some others, we considered them all equally important. To this end, we determined the rank of miRNAs for each measure of variability and each of these two normalization strategies: the least variable miR-NAs were ranked 1, the next least variable miRNAs were ranked 2 and so on.

Additionally, for the 'pooled' group, we compared the mean of the logarithms of five least variable miRNAs based on the sum of the ranks and the logarithms of five next least variable miRNAs. (Recall that the mean of the logarithms is equal to the logarithm of the geometric mean.) The comparison was performed using the paired Grambsch test which tests difference between variances, and the paired Bonett-Seier test which tests difference between mean absolute deviations from median.

Next, we compared the results of the different normalization strategies within the 'pooled' group. We compared the ranking orders of five best miRNAs for each pair of normalization strategies. Five best miRNAs were selected as the miRNAs with the least variances. The comparison was performed using the following permutation test. For each pair of normalization strategies, five best miRNAs were put on a list. The list could have less than ten entries, if some of the five best miRNAs were indicated by both normalization strategies. Next, each miRNA was ranked (that is, was assigned its serial number in the sequence arranged in ascending order of variance) for each normalization strategy. The ranks of the miRNA in this list were compared between two normalization strategies using the permutation test. The p value was the probability of obtaining a permutation "as or more extreme" than the permutation corre-sponding to the difference between these two normalization strategies, assuming all the per-mutations are of equal probability. "As or more extreme" meant that the sum of the absolute differences between the ranks is not less than the observed sum of the absolute differences between the ranks for this pair of normalization strategies. (For each miRNA, the absolute dif-ference of its ranks under two normalization strategies is found and then all the absolute differ-ences so found are summed for all miRNAs in the considered sequence.)

## Results

Table 1 presents ten most stable miRNAs in the samples: miR-361-3p, miR-151a-3p and miR-29b-3p in thyroid tissue; miR-140-5p, miR-148b-3p and miR-362-5p in bone marrow tissue; and miR-15a-5p, miR-194-5p and miR-532-5p in brain gliomas and surrounding normal tis-sue. Additionally, miR-423-5p, miR-28-5p and miR-532-5p are the most stable miRNAs in any of these tissues.

For all patients, we compared the mean of the logarithms of the concentrations of five least variable miRNAs based on the sum of the ranks and the mean of the logarithms of five next least variable miRNAs. The results of the comparison are presented in S4–S7 Tables. When we normalized miRNAs to housekeeping genes, p was less than 0.001 for both the paired

**Table 1. MicroRNAs with the most stable expression in different tissues according to NanoString data.**

| All tumors | | Thyroid gland | | Bone marrow | | Gliomas | |
|---|---|---|---|---|---|---|---|
| Best variables by combined rank | Rank sum | Most stable miRNAs by combined rank | Rank sum | Most stable miRNAs by combined rank | Rank sum | Most stable miRNAs by combined rank | Rank sum |
| miR-423-5p | 64 | miR-361-3p | 27 | miR-140-5p | 77 | miR-15a-5p | 125 |
| miR-28-5p | 65 | miR-151a-3p | 122 | miR-148b-3p | 112 | miR-194-5p | 153 |
| miR-532-5p | 66 | miR-29b-3p | 126 | miR-362-5p | 143 | miR-532-5p | 155 |
| miR-362-5p | 70 | miR-425-5p | 129 | miR-191-5p | 153 | miR-500a-5p + miR-501-5p | 173 |
| miR-425-5p | 72 | miR-361-5p | 135 | miR-378i | 154 | miR-660-5p | 193 |
| miR-361-3p | 110 | let-7b-5p | 137 | miR-98-5p | 174 | miR-185-5p | 201 |
| miR-191-5p | 110 | miR-423-5p | 145 | miR-374b-5p | 192 | miR-99b-5p | 234 |
| miR-140-5p | 114 | miR-330-3p | 154 | miR-423-5p | 194 | let-7d-5p | 235 |
| let-7a-5p | 125 | miR-28-5p | 154 | miR-29b-3p | 215 | miR-19a-3p | 249 |
| miR-106b-5p | 131 | miR-301a-3p | 169 | miR-107 | 227 | miR-140-3p | 251 |

Grambsch test and the paired Bonett-Seier test, which confirmed the assumption of different variances (the paired Grambsch test) and the assumption of different mean absolute deviations from the median (the paired Bonett-Seier test). However, when we normalized the same miR-NAs to the total miRNA content, p was equal to 0.13 for the paired Grambsch test and 0.70 for the paired Bonett-Seier test.

We performed six pair-wise comparisons of four normalization strategies with the permutation test applied to five (for each group) microRNAs with least variances (for results, see Table 2). Each p-value is reported as adjusted p-value and unadjusted p-value. The Benjamini-Hochberg adjustment was used.

As we can see, the different normalization strategies lead to different 'least variable' miR-NAs. The difference between the normalization to the total miRNA content and the normalization to 75 most highly expressed miRNAs failed to reach significance, as was expected.

## Discussion

MicroRNAs are biomarkers, which show promise for diagnosis of tumors of various origins, assessment of antitumor therapy efficacy, survival prognosis and appear to be potential targets for new antitumor drugs [6]. The most frequently used technique for analysis of miRNA expression levels is real-time RT-PCR [21]. The main problem with qPCR-based protocols is one of the choice of an optimum reference gene.

To quantify miRNAs, researchers often use the NanoString system, which directly counts RNA or DNA molecules without amplification, thus preventing PCR-related errors [20]. In this work, we used NanoString to quantify miRNAs in three tissue types: tumoral and normal

**Table 2. A comparison of various normalization strategies.**

| Normalization to | Adjusted p-value (unadjusted p-value) | | | |
|---|---|---|---|---|
| | Housekeeping genes | Total miRNA content | 75 most highly expressed miRNAs | Positive control |
| Housekeeping genes | 1 | 0.195 (0.13) | **0.028 (0.014)** | 0.77 (0.77) |
| Total miRNA content | 0.195 (0.13) | 1 | 0.77 (0.77) | **0.028 (0.014)** |
| 75 most highly expressed miRNAs | **0.028 (0.014)** | 0.77 (0.77) | 1 | **0.028 (0.014)** |
| Positive control | 0.77 (0.77) | **0.028 (0.014)** | **0.028 (0.014)** | 1 |

Statistically significant differences are in bold (adjusted p < 0.05).

thyroid tissue; brain gliomas and adjacent morphologically normal tissue; and bone marrow tissue with tumoral and non-malignant bone marrow pathologies.

NanoString detect the expression of up to 800 miRNAs in a sample. Normally, only about 200 miRNAs are expressed at sufficiently high levels, with some of them varying between the samples and the others being invariant. The authors of all publications about NanoString-based miRNA quantification data are largely interested in finding the most significant differences in miRNA expression levels between the groups [22, 23]. By contrast, we wanted to find those miRNAs that have stable expression in all groups. These miRNAs could further be used as reference genes in an analysis of miRNA expression levels by real-time RT-PCR.

We found that ten most stable miRNAs in the thyroid gland are (in descending order) miR-361-3p, -151a-3p, -29b-3p, -425-5p, -361-5p, let-7b-5p, -423-5p -330-3p, -28-5p and -301a-3p. Therefore, any alone or their combinations may be used as a reference gene for analysis of miRNA expression levels in the thyroid gland by real-time RT-PCR.

In practice, analyses of thyroid tissue by qPCR most frequently involve U6, U44 or U48 as reference genes [24–26]. However, some authors prefer miRNAs. Mohamad Yusof et al. used miR-10b-5p and miR-191-5p [27]. Santos et al. proposed a combination of let-7a, miR-103, miR-125a-5p, let-7b, miR-145 and RNU48 for work with cytological specimens of thyroid tissue [28]. Titov et al. used the geometric mean of miR-197, -99a, -151a and -214 for work with cytological specimens of FNA-biopsied thyroid material and, to make their choice of reference genes, relied on NanoString-based miRNA counts [29]. As can be seen, only miRNA-151a appears in that and the present study. This could be due to the difference in the number of study specimens: Titov et al. used 12 and we used 36. Apparently, an increase in sample size may have led to a shift in statistically significant results. However, in a later work, Titov et al. used the geometric mean of miR-197-3p, -23a-3p, and -29b-3p for work with the same material [30]. In the present work, we found one them, miRNA-29b, to be the most adequate reference gene for analysis of miRNA expression levels in thyroid tissue.

We found that ten most stable miRNAs in brain tissue are (in descending order) miR-15a-5p, -194-5p, -532-5p, -500a-5p + -501-5p, -660-5p, -185-5p, -99b-5p, let-7d-5p, -19a-3p and -140-3p. Analysis of literature data showed that U6, U48 or a combination of several small nuclear RNAs are most frequently used as reference genes [31–34]. Thus, the results of this work allow a complex normalizer to be formed, consisting of the geometric mean of miRNAs stably expressed in brain tissue. The use of a complex normalizing factor as a reference gene will increase the accuracy of analysis of miRNA expression variation.

We found that ten most stable miRNAs in the bone marrow were (in descending order) miR-140-5p, -148b-3p, -362-5p, -191-5p, -378i, -98-5p, -374b-5p, -423-5p, -29b-3p and -107. In practice, the researchers tend to give more preference to the use of small nuclear RNAs as reference genes for qPCR in bone marrow tissues [35–37]. However, some authors are confident with miRNAs. Morenos et al. used a combination of miR-16 and miR-26b for work with archived bone marrow specimens [38]. Drobna et al. used a combination of miR-16-5p, -25-3p and let-7a-5p for work with bone marrow samples from patients with acute leukemia [39]. Costé et al. used miR-191-5p for analysis of miRNA expression levels in human multipotent stromal cells [40]. Kovynev et al. used a combination of miR-103a, -191 and -378 for work with bone marrow samples from AML and ALL patients [41]. In our previous work, when we were deciding on the reference gene for analysis of miRNA expression levels in bone marrow samples, we used NanoString nCounter and chose the geometric mean of the expression levels of miR-378 and miR-191. Analysis of miRNA quantification data obtained with NanoString nCounter showed that the abundance of these miRNAs in the samples was well above background values, but these miRNAs had the least variable expression across the groups [22].

In addition to the list of tissue-specific reference genes, we composed a list of miRNAs that can be used as reference genes, no matter which tissue type they were expressed in (10 most stable, in descending order): miR-423-5p, -28-5p, -532-5p, -362-5p, -425-5p, -361-3p, -191-5p, -140-5p, let-7a-5p and -106b-5p.

In this work, miRNA-423 has the least variable expression across the study tissue types, and so it is the best candidate for a universal reference gene. To date, several works with miRNA-423 used as a reference gene have been published. Costa-Pinheiro et al. opine that miR-423-5p is an optimal reference gene for analysis of prostate cancer tissue [42]. Yanokura et al. used miRNA-423-5p as a reference gene for analysis of endometrial cancer tissue [43]. Babion et al. used miRNA-423-5p for analysis of cervical cancer tissue [44].

Analysis of literature data shows small nuclear RNAs are most frequently used as reference genes; however, their expression stability varies. Masè et al. demonstrated that, in atrial tissue samples, the most stable reference gene was SNORD48 and the least stable reference gene was U6 [45]. However, miRNAs are now being used as reference genes increasingly frequently. Remarkably, some of the miRNAs that we have chosen had already been used as reference genes for analysis of miRNA expression levels in tumors of various origins. In particular, Peltier et al., who used geNorm and NormFinder for choosing an optimal reference gene, showed that miRNA-191 and miRNA-103 were the most stable RNAs no matter which tissue type or which storage conditions (frozen specimens/FFPE) [46]. In studies about miRNA expression in renal cell adenocarcinoma, miR-28, -103, -106a and RNU48 were found to be the most stably expressed genes. If it is possible to use a single gene as a reference gene, miR-28 is recommended; otherwise a combination of miR-28 and -103 or a combination of miR-28, -103 and -106a is preferable [47]. Shen et al. and Leitão Mda et al. used a combination of miR-23a and miR-191 as a reference gene for analysis of cervical cancer tissue [48, 49]. When working with lung cancer tissue, a combination of miR-26a, -140-5p, -195 and -30b can be an option [50]. Zhang et al. showed that it is possible to use combinations of miRNA-191 and -103 as a reference gene for analysis of lung cancer FFPE tissue samples [51]. Fochi et al.' choice of reference gene in melanoma cells was miR-191-5p [52]. Zhu et al. proposed a combination of miR-103 and miR-191 for analysis of hepatocellular carcinoma FFPE samples [53]. Rohan et al. chose miR-191 and RNU6b as reference genes for analysis of cervical cancer FFPE samples [54]. Bignotti et al. used miRNA -191-5p for analysis of ovarian cancer samples [55]. Anauate et al. demonstrated that a combination of miR-101-3p and miR-140-3p was the best reference gene for analysis of stomach cancer samples [56]. As a reference gene, Jacobsen et al. use a combination of miR-24-3p, miR-151a-5p and miR-425-5p for analysis of hepatocytes [12].

Noteworthy, the miRNA most frequently used as a reference gene in work with tumors of various origins is miRNA-191. It is possible that this could is a promising universal reference gene for analysis of miRNA gene expression. However, about two hundred published works indicate that miRNA-191 is associated with various diseases [57]. This is a reminder that a particular reference gene -either as a single miRNA or, to keep the effect of the variable expression of the reference gene on the results to a minimum, as a combination of miRNAs—should be chosen for each experimental system.

To date, a large number of works attempting to find both tissue-specific and universal reference genes have been published, but the problem still persists. Research to uncover the role of miRNA in various diseases has been under way for 20 years. To be sure, much progress has been made in the development of qPCR data normalization strategies in an analysis of miRNA expression levels–from choosing suitable small nuclear RNAs to the development of mathematical approaches minimizing the interference of a reference gene with the end result. In the current work, we have identified the most stable miRNAs by a NanoString-enabled analysis of miRNA expression levels: miR-361-3p, -151a-3p and -29b-3p for thyroid tissue; miR-15a-5p,

-194-5p and -532-5p for brain tissue; miR-140-5p, -148b-3p and -362-5p for bone marrow tissue; and miR-423-5p, -28-5p and -532-5p for any of the tissues. The study included very few tissue samples, but even so results of several comparisons were statistically significant. It is deemed logical to further validate these results on larger sample sizes using qPCR followed by analysis of the most stable miRNAs by GeNorm, NormFinder or Best Keeper.

## Supporting information

**S1 Table. Patient characteristics at the time of diagnosis (thyroid tissue).**
(DOCX)

**S2 Table. Patient characteristics at the time of diagnosis (brain tissue).**
(DOCX)

**S3 Table. Patient characteristics at the time of diagnosis (bone marrow tissue).**
(DOCX)

**S4 Table. MicroRNAs with the least variable expression in all groups of patients.** The names of the microRNA's together with the observed values of deviations are reported.
(DOCX)

**S5 Table. MicroRNAs with the least variable expression in thyroid tissue.** The names of the miRNAs together with the observed values of deviations are reported. If the names of two miRNAs are linked with the plus sign "+", then only the total content of these two miRNAs was measured.
(DOCX)

**S6 Table. MicroRNAs with the least variable expression in bone marrow tissue.** The names of the miRNAs together with the observed values of deviations are reported. If the names of two miRNAs are linked with the plus sign "+", then only the total content of these two miRNAs was measured.
(DOCX)

**S7 Table. MicroRNAs with the least variable expression in brain tissue.** The names of the miRNAs together with the observed values of deviations are reported. If the names of two miRNAs are linked with the plus sign "+", then only the total content of these two miRNAs was measured.
(DOCX)

## Author Contributions

**Conceptualization:** Yuliya A. Veryaskina, Sergei E. Titov.

**Formal analysis:** Yuliya A. Veryaskina, Sergei E. Titov, Pavel S. Ruzankin, Anton S. Tarasenko.

**Methodology:** Sergei E. Titov.

**Resources:** Sergei P. Shevchenko, Igor B. Kovynev, Evgenij V. Stupak.

**Software:** Pavel S. Ruzankin, Anton S. Tarasenko.

**Supervision:** Mikhail K. Ivanov, Tatiana I. Pospelova, Igor F. Zhimulev.

**Writing – original draft:** Yuliya A. Veryaskina, Pavel S. Ruzankin.

**Writing – review & editing:** Sergei E. Titov.

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
