## [Decision Letter · Decision Letter 0]

15 Oct 2021

PONE-D-21-19542Selection of reference genes for quantitative analysis of microRNA expression in three cancers.PLOS ONE

Dear Dr. Veryaskina,

Thank you for submitting your manuscript to PLOS ONE. After careful consideration, we feel that it has merit but does not fully meet PLOS ONE’s publication criteria as it currently stands. Therefore, we invite you to submit a revised version of the manuscript that addresses the points raised during the review process.

We look forward to receiving your revised manuscript.

Kind regards,

Gayle E. Woloschak, PhD

Academic Editor

PLOS ONE

Journal Requirements:

2. One of the Author is affiliated to AO Vector-Best. This information should be added to the COI/FD statement.

4. Thank you for stating the following in the Acknowledgments / Funding Section of your manuscript: 

The work of S. Titov was financially supported by the Russian Science Foundation (project No. 20-14-00074). The work of Y. Veryaskina was supported by the Russian Foundation for Basic Research (project No. 19-34-60024). The work of P. Ruzankin and A. Tarasenko was supported by the Mathematical Center in Akademgorodok under agreement No. 075-15-2019-1675 with the Ministry of Science and Higher Education of the Russian Federation. 

The work of S. Titov was financially supported by the Russian Science Foun-dation (project No. 20-14-00074). The work of Y. Veryaskina was supported by the Russian Foundation for Basic Research (project No. 19-34-60024). The work of P. Ruzankin and A. Tarasenko was supported by the Mathematical Center in Akademgorodok under agreement No. 075-15-2019-1675 with the Ministry of Science and Higher Education of the Russian Federation. 

 The work of S. Titov was financially supported by the Russian Science Foun-dation (project No. 20-14-00074). The work of Y. Veryaskina was supported by the Russian Foundation for Basic Research (project No. 19-34-60024). The work of P. Ruzankin and A. Tarasenko was supported by the Mathematical Center in Akademgorodok under agreement No. 075-15-2019-1675 with the Ministry of Science and Higher Education of the Russian Federation. 

Additional Editor Comments:

Some minor changes were noted by both reviewers. Please address these in a revision.

Reviewers' comments:

Reviewer's Responses to Questions

**Comments to the Author**

1. Is the manuscript technically sound, and do the data support the conclusions?

Reviewer #1: Yes

Reviewer #2: Yes

2. Has the statistical analysis been performed appropriately and rigorously? 

Reviewer #1: Yes

Reviewer #2: Yes

3. Have the authors made all data underlying the findings in their manuscript fully available?

Reviewer #1: Yes

Reviewer #2: Yes

4. Is the manuscript presented in an intelligible fashion and written in standard English?

Reviewer #1: Yes

Reviewer #2: Yes

5. Review Comments to the Author

Reviewer #1: The aim of this article was to find the miRNAs with stable expression in the thyroid gland, brain and bone marrow cancer and normal tissues. These miRNAs supposed to be suitable as endogenous controls for miRNAs quantification experiments. RT-qPCR requires endogenous controls for result normalization and reliability. The endogenous control helps to correct differences between sample quality and variations during RNA extraction or reverse transcription procedures. Housekeeping genes, ribosomal, small nuclear or nucleolar RNAs can play the role of such internal controls. However, expression levels of these genes may differ in neoplastic and normal tissues. To find proper endogenous control, the authors evaluated expression stability about of 800 miRNAs with help of NanoString nCounter Assay for miRNAs quantification. Their recommendations are to use as internal controls the following miRNAs: miR-361-3p, -151a-3p and -29b- 3p for thyroid gland; miR-15a-5p, -194-5p and -532-5p for brain; miR-140-5p, -148b- 3p and -362-5p for bone marrow; and miR-423-5p, -28-5p and -532-5p as the universal controls for all three tissue types. I suppose this important paper will be useful for the future investigations in the field of RT-qPCR-based miRNAs quantification. But, I have several questions for the authors:

1. On the page 5, lines 116-117, the authors wrote : “the control group consisted of patients with non-cancerous blood diseases”. What was the reason to use these patients as the control group? Because different diseases, even non-cancerous, can significantly change miRNAs expression profile in patients tissues in compare with healthy ones.

2. Would the authors kindly provide a list of housekeeping genes and positive controls, used for normalization?

3. Let-7a, -7b and -7d were listed in Table 1 among a least variable miRNAs in thyroid gland, brain and as universal controls correspondently. But, miRNAs from let-7 family are well known tumor suppressors. Expression of these miRs is downregulated in tumors. How can it be possible that these miRNAs are the most stable?

4. There is no confidence interval for p-values in the Table 2. Significant p-values in this table may be shown in bold or color.

5. MiR-28-5p and miR-532-5p were chosen like the universal controls for all three tissues. But, among 10 least variable miRNAs, miR-28-5p is only ninth in thyroid gland and even was not listed in Table 1 for brain and bone marrow. The same story is for miR-532-5p. This miRNA is stable in brain tissues only. What is the possible reason for these discrepancies?

Reviewer #2: PONE-D-21-19542 Selection of reference genes for quantitative analysis of microRNA expression in three cancers.

The Manuscript presents expression analysis of miRNAs in three types of cancer and adjacent normal tissue by NanoString nCounter miRNA quantification data in order to find ones stably expressed across analyzed tissues. The authors emphasized that widely used qPCR technique for expression quantification needs more accurate data regarding stably expressed miRNAs for normalizations. The authors came to a conclusion that at least 3 analyzed miRNAs are expressed consistently in the selected tissues. Moreover, 3 miRNAs were found to be consistent regardless the tissue type, which makes them good candidates for general use. The presented research is highly important because lacking adequate data regarding this matter can lead to misinterpretation of obtained results.

The Manuscript is well written, concise and should be accepted in the present form. The only thing that I would suggest is to correct the title into: Selection of reference genes for quantitative analysis of microRNA expression in three different types of cancers or even more specific in three types of cancers with thyroid, brain and bone marrow origin.

6. PLOS authors have the option to publish the peer review history of their article (what does this mean?). If published, this will include your full peer review and any attached files.

Reviewer #1: No

Reviewer #2: No

---

## [Author Response · Author response to Decision Letter 0]

15 Dec 2021

We thank the Reviewers for their insightful comments. Below are our answers to the Reviewers’ comments shown in bold.

Reviewer #1

1. On the page 5, lines 116-117, the authors wrote: “the control group consisted of patients with non-cancerous blood diseases”. What was the reason to use these patients as the control group? Because different diseases, even non-cancerous, can significantly change miRNAs expression profile in patients’ tissues in compare with healthy ones. 

We have added the following explanation on p.5

“Work with healthy bone marrow donors for allogeneic transplantation is beyond the competence of our clinic. Taken together, we decided to go with a control group composed of people who had no hematologic cancer, but had indications for bone marrow examination to exclude one. They were people with secondary anemic and cytopenic conditions, in whom leukemias were not confirmed by myelography.”

2. Would the authors kindly provide a list of housekeeping genes and positive controls, used for normalization? 

To normalize the data, nCounter technology used: 

1) Positive Controls (n=6). Probes that recognize synthetic mRNA targets included in the CodeSet at specified concentrations (targets do not require ligation). Positive controls used by the QC metrics in nSolver to confirm linear response to input amounts, and confirm that low input signal is above background.

2) Housekeeping genes: ACTB, B2M, GAPDH, RPL19, RPLP0.

We have added the following explanation on p.6

3. Let-7a, -7b and -7d were listed in Table 1 among a least variable miRNAs in thyroid gland, brain and as universal controls correspondently. But, miRNAs from let-7 family are well known tumor suppressors. Expression of these miRs is downregulated in tumors. How can it be possible that these miRNAs are the most stable?

Analysis of the literature revealed that expression levels of members of the Let-7 family were found significantly different between tumor and normal tissue samples in several publications. The value of the difference varies both over different tissues and within individual tissues. The following ratios of concentrations were reported: Thyroid tumors (cancer sample versus normal): 0.61 (hsa-let-7d-5p) (Swierniak et al,2013); 1.21 (has-let-7f-1) (Pallante et al,2006); approximately 1.5 (has-let-7d) and approximately 1 (has-let-7g) (Braun et al,2010); approximately 3 (has-let-7a) (Zhou et al, 2017). Glioma (cancer sample versus normal): 1.69 (has-let-7b) (Zhang et al,2019), approximately 2 (has-let-7d and has-let-7a) (Yang et al). Bone marrow (cancer sample versus normal): (1.35-3.78) (Veryaskina et al,2020). We see that, in most of the cases, the difference between the expression levels was about 2 or less. In most of the studies, no correction for multiple comparisons was employed, which could lead to chance findings. Besides, generally, authors tend to mention more frequently miRNAs with significant differences than those for which comparisons yielded negative results. Therefore, it seems impossible to evaluate the number of studies where the differences for the Let-7 family miRNAs were not found significant.

On the other hand, only a small number of samples were used in our work. Therefore, the results are to be validated in succeeding studies.

 It is also worth noting that let-7a, let-7b, and let-7d were not the most stable normalizers in our study.

4. There is no confidence interval for p-values in the Table 2. Significant p-values in this table may be shown in bold or color. 

We might misunderstand the comment. As far as we know, confidence intervals for p-values were suggested only in some theoretical constructs and are not used in practice. Probably, the Reviewer suggested considering confidence intervals for the differences corresponding to the p-values. For Table 2, permutation tests were used, the differences being the sums of absolute differences of the corresponding ranks, which is described in the Statistical analysis section. As far as we know, there is no straightforward way to introduce confidence intervals for differences in permutations tests. Therefore, we did not present confidence intervals in Table 2.

The significant p-values are now shown in bold in Table 2.

5. MiR-28-5p and miR-532-5p were chosen like the universal controls for all three tissues. But, among 10 least variable miRNAs, miR-28-5p is only ninth in thyroid gland and even was not listed in Table 1 for brain and bone marrow. The same story is for miR-532-5p. This miRNA is stable in brain tissues only. What is the possible reason for these discrepancies? 

For a miRNA, the variability in all tumors depends on within-group variability in each group and between-the-groups variability of the mean level of concentration of the miRNA. A miRNA may have low within-group variability in the groups, but high between-the-groups variability of the mean level of concentration; or vice versa. Therefore, the order of the miRNAs for all tumors may not correlate with the orders for individual tumors.

Reviewer #2

The only thing that I would suggest is to correct the title into: Selection of reference genes for quantitative analysis of microRNA expression in three different types of cancers or even more specific in three types of cancers with thyroid, brain and bone marrow origin.

According to the Reviewer’s suggestion, the title is corrected into: Selection of reference genes for quantitative analysis of microRNA expression in three different types of cancer.

---

## [Decision Letter · Decision Letter 1]

31 Jan 2022

Selection of reference genes for quantitative analysis of microRNA expression in three different types of cancer.

PONE-D-21-19542R1

Dear Dr. Veryaskina:

We’re pleased to inform you that your manuscript has been judged scientifically suitable for publication and will be formally accepted for publication once it meets all outstanding technical requirements.

Kind regards,

Gayle E. Woloschak, PhD

Section Editor

PLOS ONE

Additional Editor Comments (optional):

Thank you for addressing the concerns raised by the reviewers.

Reviewers' comments:

Reviewer's Responses to Questions

**Comments to the Author**

1. If the authors have adequately addressed your comments raised in a previous round of review and you feel that this manuscript is now acceptable for publication, you may indicate that here to bypass the “Comments to the Author” section, enter your conflict of interest statement in the “Confidential to Editor” section, and submit your "Accept" recommendation.

Reviewer #1: (No Response)

Reviewer #2: All comments have been addressed

2. Is the manuscript technically sound, and do the data support the conclusions?

Reviewer #1: Yes

Reviewer #2: Yes

3. Has the statistical analysis been performed appropriately and rigorously? 

Reviewer #1: Yes

Reviewer #2: I Don't Know

4. Have the authors made all data underlying the findings in their manuscript fully available?

Reviewer #1: Yes

Reviewer #2: Yes

5. Is the manuscript presented in an intelligible fashion and written in standard English?

Reviewer #1: Yes

Reviewer #2: Yes

6. Review Comments to the Author

Reviewer #1: (No Response)

Reviewer #2: (No Response)

7. PLOS authors have the option to publish the peer review history of their article (what does this mean?). If published, this will include your full peer review and any attached files.

Reviewer #1: No

Reviewer #2: No

---

## [Editor Report · Acceptance letter]

8 Feb 2022

PONE-D-21-19542R1 

Selection of reference genes for quantitative analysis of microRNA expression in three different types of cancer. 

Dear Dr. Veryaskina:

I'm pleased to inform you that your manuscript has been deemed suitable for publication in PLOS ONE. Congratulations! Your manuscript is now with our production department. 

Kind regards, 

on behalf of

Dr. Gayle E. Woloschak 

Section Editor

PLOS ONE